# Exploring medical error taxonomies and human factors in simulation-based healthcare education

**Tamara Skrisovska**[1,2]☯, **Daniel Schwarz**[1]☯\*, **Martina Kosinova**[1,2]‡, **Petr Stourac**[1,2]‡

**1** Faculty of Medicine, Department of Simulation Medicine, Masaryk University, Brno, Czech Republic,
**2** University Hospital Brno and Faculty of Medicine, Department of Pediatric Anesthesiology and Intensive Care Medicine, Masaryk University, Brno, Czech Republic

☯ These authors contributed equally to this work.
‡ These authors also contributed equally to this work.
\* schwarz@med.muni.cz

**Data Availability Statement:** All relevant data for this study are publicly available from the Zenodo repository (https://doi.org/10.5281/zenodo.14591686).

## Abstract

This study aims to provide an updated overview of medical error taxonomies by building on a robust review conducted in 2011. It seeks to identify the key characteristics of the most suitable taxonomy for use in high-fidelity simulation-based postgraduate courses in Critical Care. While many taxonomies are available, none seem to be explicitly designed for the unique context of healthcare simulation-based education, in which errors are regarded as essential learning opportunities. Rather than creating a new classification system, this study proposes integrating existing taxonomies to enhance their applicability in simulation training. Through data from surveys of participants and tutors in postgraduate simulation-based courses, this study provides an exploratory analysis of whether a generic or domain-specific taxonomy is more suitable for healthcare education. While a generic classification may cover a broad spectrum of errors, a domain-specific approach could be more relatable and practical for healthcare professionals in a given domain, potentially improving error-reporting rates. Seven strong links were identified in the reviewed classification systems. These correlations allowed the authors to propose various simulation training strategies to address the errors identified in both the classification systems. This approach focuses on error management and fostering a safety culture, aiming to reduce communication-related errors by introducing the principles of Crisis Resource Management, effective communication methods, and overall teamwork improvement. The gathered data contributes to a better understanding and training of the most prevalent medical errors, with significant correlations found between different medical error taxonomies, suggesting that addressing one can positively impact others. The study highlights the importance of simulation-based education in healthcare for error management and analysis.

## Introduction

Medical error is an act of omission or commission during planning or execution that contributes to or could contribute to unintended harm to a patient [1]. Data show that medical errors

**Funding:** This research was partially supported by the Specific University Research grant provided to Masaryk University by the Ministry of Education of the Czech Republic (MUNI/A/1595/2023, MUNI/A/1551/2023) and also in part supported by the Ministry of Health of the Czech Republic (FNBr, 65269705).

**Competing interests:** The authors have declared that no competing interests exist.

are among the top three causes of death worldwide, as highlighted by pivotal studies [2,3]. Recognizing the severity of the situation, the World Health Organization (WHO) has made patient safety a global health priority with its "Global Action on Patient Safety: A Decade of Patient Safety 2020–2030" initiative [4]. This underscores that unsafe patient care remains a critical global issue that places a heavy burden on healthcare systems. Given the complexity of healthcare systems and the involvement of human factors, achieving an error-free system is not feasible, and the focus is on minimizing the occurrence and impact of medical errors [5,6]. Key aspects for enhancing patient safety involve not only understanding and analyzing the causes of errors at both individual and systemic levels but also establishing resilient systems that can respond to, recover from, and adapt to these errors [7,8]. Achieving resilience requires robust error reporting, which fundamentally depends on standard terminology and classification systems. Such a system enables consistent communication, tracking, and comparison of incidents, thereby forming the foundation of a structured framework for medical error analysis [9,10]. A taxonomy, which is an organized, hierarchical classification system, is particularly valuable in this context as it allows for a detailed understanding of the relationships between error categories [11,12]. An effective medical error taxonomy should support documentation, prediction, and error reduction, all of which are integral to meaningful reporting [13]. Using consistent language and standard terminology enables the coding, filtering, sorting, and organizing of error data [14], making it suitable for both individual and systemic analyses and interventions. Different types of errors require tailored intervention strategies, and a well-structured taxonomy can guide these approaches, thereby strengthening the overall resilience of healthcare practices (Fig 1). The terms "error taxonomy" and "error classification systems" are often used interchangeably because they organize and categorize errors for better understanding and management. However, some error classification systems can overwhelm healthcare providers, leading to potential underreporting or incorrect categorization, which may complicate the analysis of such reports [15].

Medical error taxonomies can be generic or domain-specific. Generic taxonomies, such as Rasmussen's (1987) classification of errors into skill-, rule-, and knowledge-based types (SRK) [16], offer broad terminology applicable across various fields, including high-reliability organizations such as aviation and nuclear power [17]. Reason's (1990) model builds upon Rasmussen's work through the Generic Error-Modelling System (GEMS), which incorporates diverse error mechanisms such as slips, lapses, and mistakes within the SRK framework. Another prominent generic taxonomy, the Human Factors Analysis and Classification System

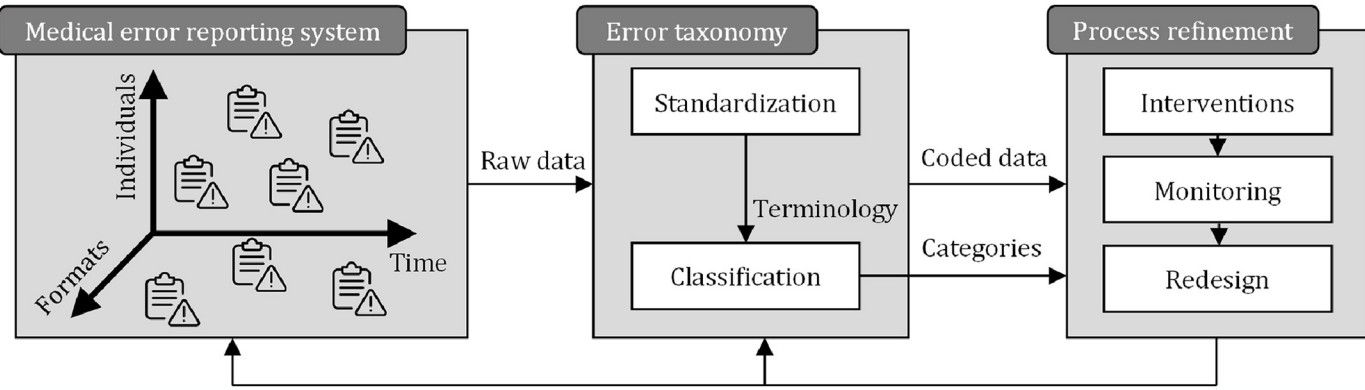

**Fig 1. The process of medical error reporting and classification as a foundation for enhancing healthcare system resilience and safety.**

(HFACS), is based on Reason's concepts but was recently found to be inadequate for analyzing adverse medical events. A new framework has since been validated for root cause analysis and the recording of such events, tailored more closely to healthcare contexts. In addition, the Joint Commission on Accreditation of Healthcare Organizations (JCAHO) Patient Safety Event Taxonomy (2005) organizes reporting systems into five main classifications: impact, type, domain, cause, and prevention and mitigation [10]. Although these generic taxonomies cover a wide range of domains, their broad generalization may limit their effectiveness in supporting patient safety recommendations within specific fields [18,19].

Conversely, domain-specific medical error taxonomies, which contain terms unique to their respective domains, display enhanced reliability because of the consistent terminology used in taxonomies, incident reports, and the final classification of errors. These taxonomies focus on specific areas of medicine such as primary care [20] and surgery [21,22]. Additional examples can be found in Critical Care, which includes emergency care and intensive medicine [23–26].

These studies commonly analyze actual patient safety incidents, subsequently creating new taxonomies or modifying existing ones to meet the practical needs of patient-centered care in the fast-paced settings of Emergency Departments and Intensive Care Units. However, the categorizations in these domain-specific taxonomies may pose challenges for information sharing across various healthcare fields, especially when considering differences in patient case severity and the time-sensitive nature of decision-making, and the potential for serious or even fatal outcomes due to errors in Critical Care compared to other domains [24].

To improve patient safety and the quality of care in Critical Care settings, it is essential to employ a systematic approach for identifying the root causes of medical errors. Our study addresses this need by exploring effective error-management strategies within simulation-based education. We assessed various taxonomies to categorize medical errors in Critical Care. Although creating a new taxonomy is beyond the scope of our study, this exploratory research offers a starting point for future efforts aimed at establishing a taxonomy that can effectively support the understanding and reduction of medical errors in this context. Our study also builds on a comprehensive review published in 2011, providing an updated perspective on medical error taxonomies and determining the essential features of a taxonomy suitable for postgraduate training in Critical Care simulation settings.

## Methods

This exploratory analysis was supported by a literature review, building on the comprehensive work of Taib [14], who investigated 26 medical error taxonomies from a human factor perspective up to 2008. Using the same methodology and keywords (patient safety, medical errors, taxonomy, and classification), we searched MEDLINE, Embase, and PubMed for relevant publications from 2009 to 2024. This review aimed to assess the evolution of these taxonomies, particularly their applicability to simulation-based healthcare education, in which medical errors are viewed as learning opportunities. Two independent researchers verified the findings to ensure consistency. Our analysis revealed a knowledge gap: no existing taxonomy adequately addresses the unique needs of simulation-based medical education, particularly in domains such as Critical Care. Such taxonomy could enhance the analysis of prevalent errors, guide the design of simulation scenarios that reflect real-world complexities, and ultimately improve the effectiveness of simulation-based medical education.

We further explored this knowledge gap by comparing two established medical error taxonomies in the environment of our simulation center. The first was Reason's GEMS [25]. One potential shortcoming of this system is its limited direct applicability in healthcare. These

A)

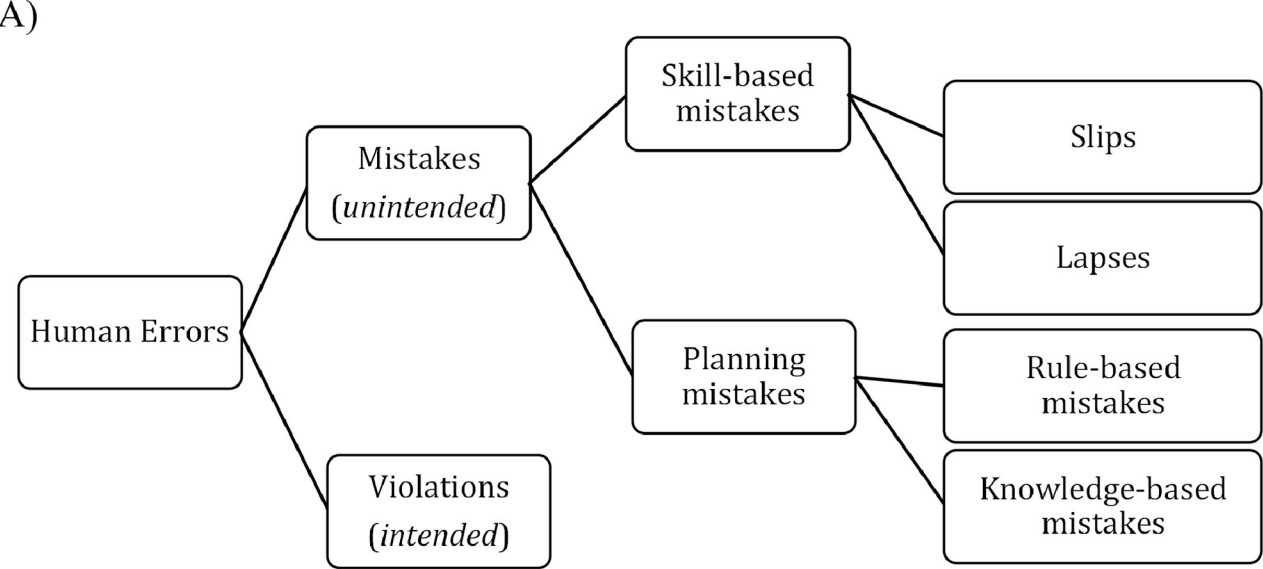

B)

| Error Category | Description |
| --- | --- |
| Violations | act knowingly incurs a risk |
| Slips | action-based errors |
| Lapses | memory-based errors |
| Rule-based mistakes | a) misapplication of a good rule or the failure to apply a good rule<br><br>b) application of a bad rule |
| Knowledge-based mistakes | related to any knowledge, general, specific, or expert |

**Fig 2.** (A) Reason's classification: Error categories. Adapted from [25]. (B) Reason's classification: Definitions.

classifications are abstract and can be applied universally, as evidenced by the scheme and error description depicted in Fig 2.

Round et al. proposed a second practical taxonomy for education sessions, which is commonly referred to as the "deadly ten errors" [26]. This taxonomy, illustrated in Fig 3, offers clinical concepts for comprehending the origins of medical errors with potential applications in various medical education settings.

We conducted a survey with postgraduate participants attending high-fidelity simulation courses in Critical Care at Masaryk University's Simulation Centre (SIMU) in Brno, the Czech Republic. The survey collected data from 124 participants and 13 tutors regarding the types and occurrence of medical errors observed or performed during the simulations, as well as their frequency in daily practice. Each simulation session was preceded by a standard briefing that included an introduction to the survey and an overview of the medical error types with

| Error Category | Description |
|---|---|
| Sloth | not doing what the effort required or perceiving as an inadequate reward |
| Fixation | particular diagnosis or analysis is firmly held onto despite evidence against it, confirmation bias |
| Communication breakdown | paramount information is not released or does not reach its destination at the right time |
| Poor team working | team members beyond their capabilities, lack of leadership |
| Playing the odds | failure to understand the fundamental rules of probability, preferring despite a well-known diagnosis |
| Bravado | clinicians work beyond their competence or lack adequate supervision |
| Ignorance | unconscious incompetence, lack of knowledge |
| Mis-triage | over – or underestimation of the seriousness of a situation and lack of prioritization |
| Lack of skill | lack of teaching or practice |
| System error | unnecessary decision-making steps, multiple distractions, lack of checklists, lack of policies |

**Fig 3. Round's classification: Descriptions.**

the definitions provided in the questionnaire (S1 Appendix). After each simulation session, structured feedback was provided to learners in the form of a debriefing, emphasizing technical and non-technical skills, with a focus on incidents that may compromise patient safety. The survey was conducted between March 1, 2021, and October 27, 2022. All participants provided informed consent at the start of the survey and the data were anonymized. The study adhered to the local laws and institutional standards and did not involve any interventions.

In our survey, we explored the occurrence and nature of medical errors in order to link observations from simulation-based courses with real-world clinical practice. Participants were asked whether they observed any type of medical error during the simulation course, with binary "yes" or "no" responses, and to specify the error types using predefined categories from both Reason's and Round's taxonomies. The participants also ranked the frequency of medical errors in their daily practice. They had the opportunity to describe the specific errors observed during the simulations and those they encountered in their own clinical experiences. This structured approach allowed for a deeper understanding of the connections between errors in simulated scenarios and those occurring in actual clinical settings, thus highlighting the educational potential of simulation-based training.

## Statistics and results

Following data collection, we examined the relationships between Reason's and Round's medical error taxonomies to identify significant associations. Fig 4 outlines the data preprocessing and analysis processes, from initial data handling to the identification of key links between these classification systems.

We further analyzed the observed occurrence matrix by determining the relative frequencies of medical errors. Fig 5 presents a stacked bar chart comparing the distribution of medical errors recorded during the simulation, as reported by both the participants and tutors.

Fig 5 shows that according to Reason's taxonomy, knowledge-based mistakes and lapses were the most common errors observed. Under Round's taxonomy, frequent instances of communication breakdown and poor team performance have been reported. In contrast, violations (Reason), a deliberate deviation from the rule or procedure, and sloths (Round), neglecting necessary actions due to the belief that the effort required outweighs the perceived benefit or reward, were among the least frequent types of errors within each taxonomy, indicating their rarity compared with other types.

To further explore the error patterns, we analyzed the ranking matrices provided by the participants and tutors. The results (Fig 6) use boxplots to visualize the rankings of medical error types, highlighting the alignment between the tutor and participant perspectives.

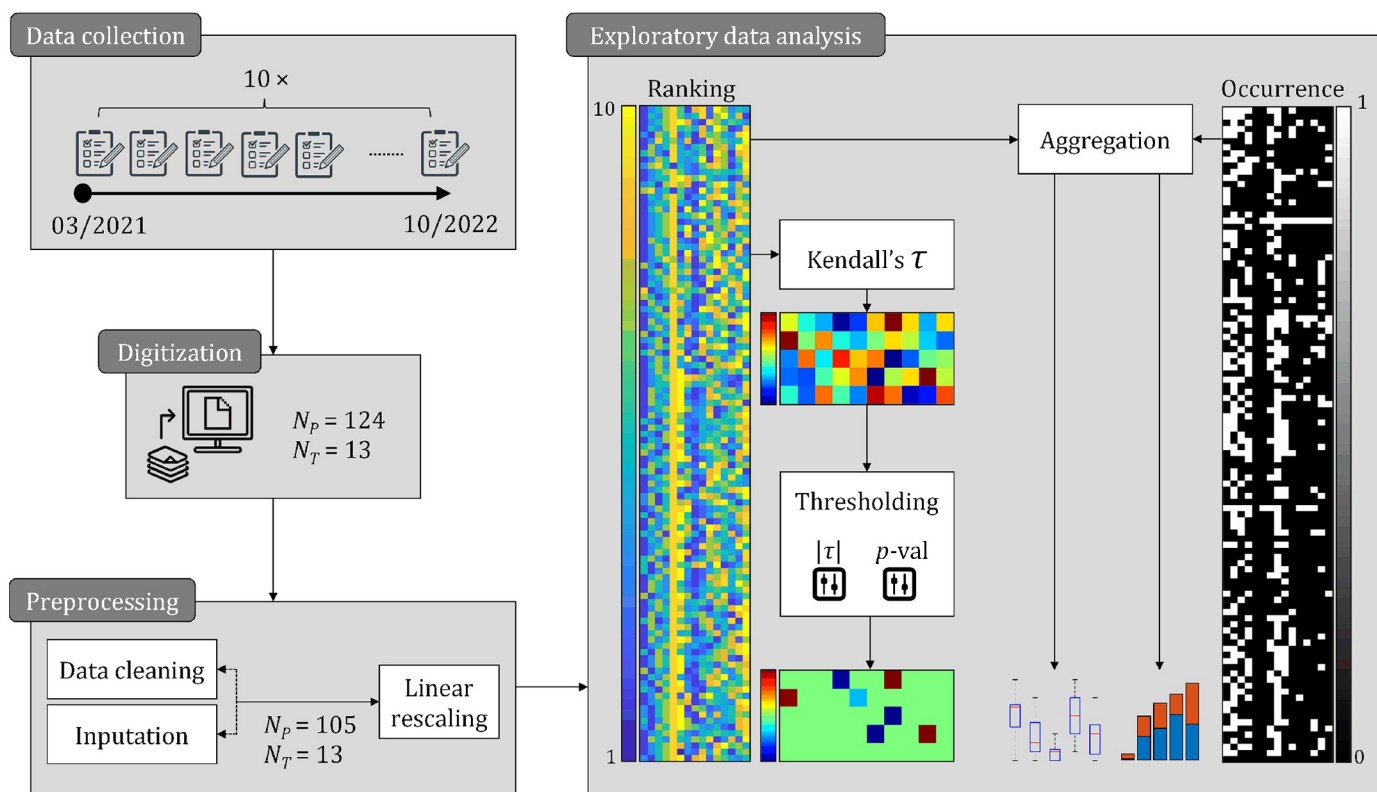

**Fig 4. Flowchart illustrating the methodology for examining medical error associations.** The methodology involved collecting data from 124 participants and 13 tutors, exploring the relationship between medical error classifications using Kendall's tau correlation coefficient, and identifying significant associations with thresholding. Data cleaning, inputting missing values, and linear rescaling of the resulting data matrix (105 × 15) were performed, followed by analysis to identify the most common types of medical errors and their associations with different classification systems.

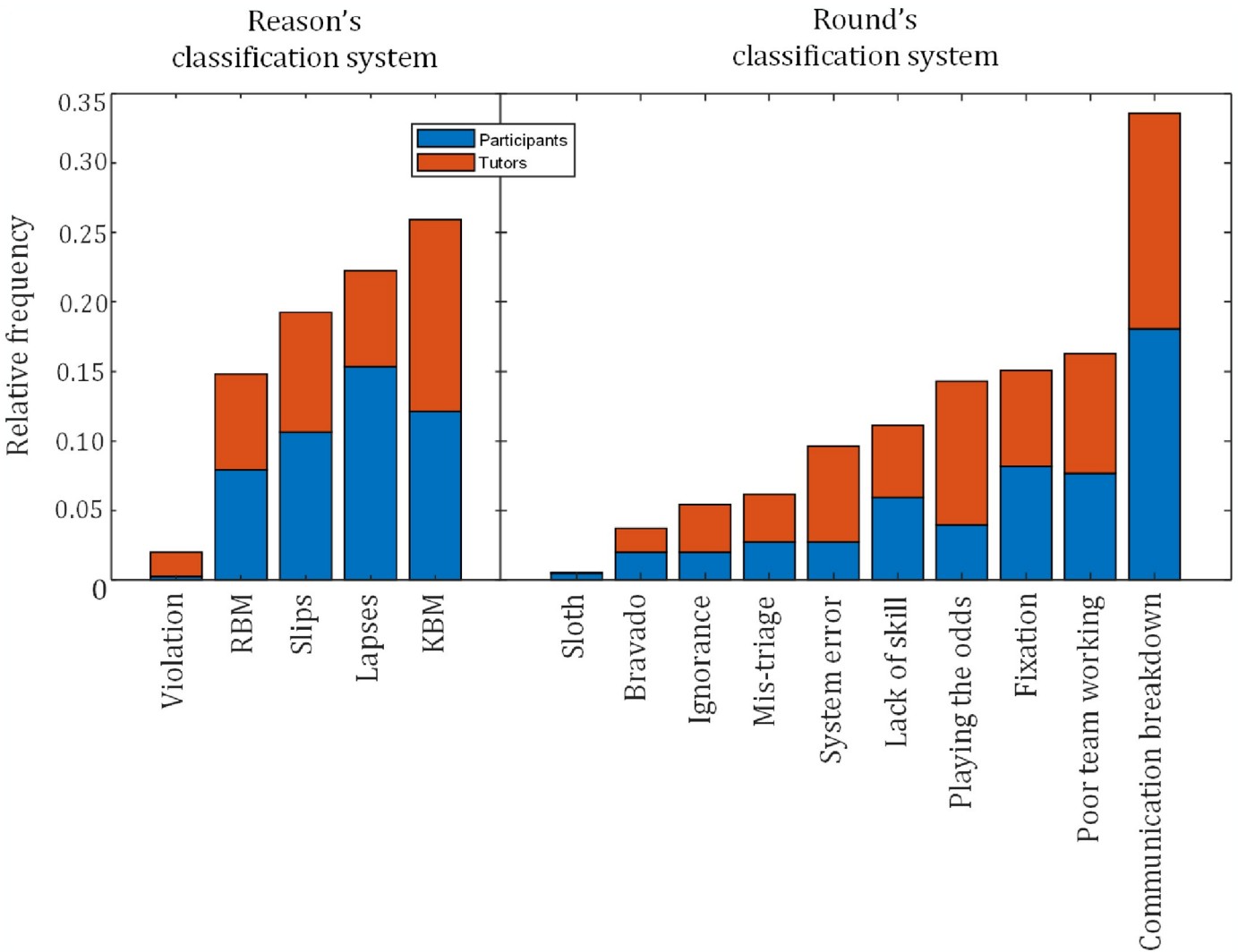

**Fig 5. Relative frequency of medical errors observed during the simulation sessions as reported by participants and tutors.** The bar charts distinguish and compare two distinct medical error classification systems—the Reason's and Round's taxonomies. The x-axis denotes the categories of medical errors, revealing the most and least frequent types of errors within each taxonomy. RBM—Rule-based Mistakes; KBM—Knowledge-based Mistakes.

Upon examining Fig 6, it is evident that the participants' and tutors' rankings were highly similar, suggesting a strong agreement in their assessment of medical error types.

We used the Kendall's tau correlation coefficient to investigate the relationship between the classification systems for medical errors by Reason and Round. Kendall's tau is a nonparametric association measure between two ranked variables, which is suitable for categorized data such as the ranked error frequencies in our study. This is ideal because Kendall's tau makes no assumptions about the data distribution and depends solely on the observation order, not the magnitude. This implies that Kendall's tau can identify variable associations even in nonlinear relationships and remains resistant to outliers.

We executed a thresholding process on the Kendall tau matrix, utilizing a p-value under 0.1 and a correlation coefficient (tau) absolute value exceeding 0.1 as thresholds. A p-value threshold of 0.1 was selected to increase sensitivity to potentially meaningful associations, acknowledging that a more relaxed threshold is appropriate for exploratory analyses like ours, where

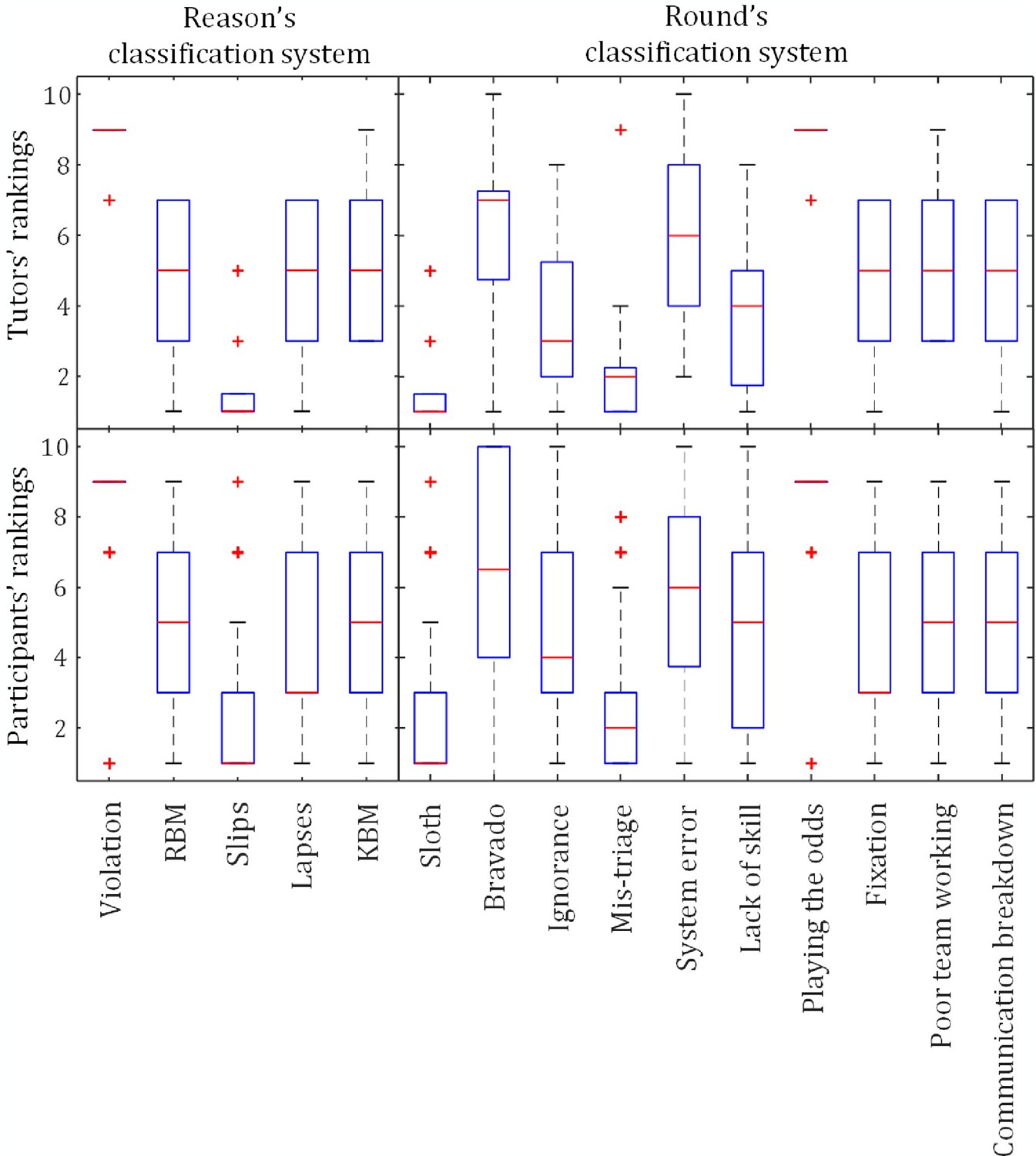

**Fig 6. Boxplot showing participants' and tutors' rankings of medical errors.** The y-axis represents the ranking, while the x-axis displays the categories of medical errors. The order of medical mistakes on the x-axis corresponds to the sorted frequencies, as shown in Fig 5. RBM—Rule-based Mistakes; KBM—Knowledge-based Mistakes.

**Table 1. Detected correlations between the two presented medical error classification systems: Reason's and Round's taxonomies.**

| Data from participants/tutors | Correlation coefficient | p-value | Reason's vs Round's classification |
|---|---|---|---|
| Participants | 0.157 | 0.044 * | Lapses vs. Playing the odds |
| Participants | -0.138 | 0.076 | RBM vs. Ignorance |
| Tutors | 0.493 | 0.047 * | KBM vs Lack of skill |
| Tutors | 0.482 | 0.048 * | Lapses vs. sloth |
| Tutors | -0.472 | 0.050 | KBM vs. Competence |
| Tutors | -0.468 | 0.074 | Slips vs Teamwork |

RBM—Rule-based Mistakes; KBM—Knowledge-based Mistakes. An asterisk (*) next to a p-value indicates a statistically significant result.

the aim is to identify trends that may warrant further investigation. Additionally, the correlation coefficient threshold of $|\tau| > 0.1$ was chosen to capture associations with at least a minimal strength of relationship, ensuring that any identified links are practically relevant while accounting for the fact that, in nonparametric data, stronger correlations are less common. This step helped us identify significant associations between various medical error taxonomies according to the frequency rankings by the participants and tutors.

Our analysis identified six significant relationships between Reason's ten error types and Round's five error types. We found two associations among participants and four among tutors, as shown in Table 1.

## Discussion

This study provides new insights into the relationships between generic and domain-specific medical error taxonomies, particularly in simulation-based medical education. By examining the correlations between Reason's and Round's classification systems, our findings underscore the significance of combining these approaches to address diverse educational and clinical needs. Using Kendall's tau correlation analysis, we identified significant relationships that highlighted the value of integrating multiple classification systems to enhance error detection and training efficacy. The proposed correlations and their potential implications illustrated in Fig 7 suggest that relying solely on a single classification system could lead to missed opportunities for identifying and addressing specific medical errors during simulation scenarios.

Our results further demonstrated the critical role of communication and teamwork in error prevention. In line with existing literature, miscommunication remains the predominant cause of medical errors in both simulation and clinical settings, often linked to issues such as unclear medication orders or insufficient verification of patient details [27,28]. These findings reaffirm that addressing communication and teamwork deficiencies is paramount for improving patient safety and outcomes.

Simulation-based education offers a unique environment for understanding and mitigating medical errors, without jeopardizing patient safety. This setting allowed the participants to experience and analyze clinical events under controlled conditions, thereby fostering the development of error management strategies. Our study emphasizes the integration of principles such as Crisis Resource Management into simulation training, which has proven to be effective in enhancing communication, situational awareness, and role clarity among healthcare teams [28,29].

Given the identified correlations between the taxonomies, we propose the development of a future pilot simulation course to explore the practicality of combining Reason's and Round's taxonomies. Although this course was not a part of the current study, its implementation

| Detected correlation | Proposed interpretation | Simulation training strategy |
|---|---|---|
| A higher frequency of lapses is associated with a higher frequency of playing-the-odds errors. | Since lapses are associated with memory and often described as 'working on autopilot', playing the odds represents the unwillingness to 'think out of the box' and approach the case from a different perspective, to admit a diagnosis other than a well-known one – correlating these two errors poses a risk to daily routine practice. | Simulation scenarios based on unusual crises (e.g. malignant hyperthermia during anaesthesia) to get participants focused on non-routine situations. |
| A higher frequency of rule-based mistakes is associated with a lower frequency of errors caused by ignorance. | RBM means not applying the correct rule or applying the wrong one for the given situation. In both cases, ignorance would suggest not using a rule due to a lack of knowledge or awareness of an existing rule. | Training with cognitive aid of the most important process flows, e.g. CPR guidelines. |
| A higher frequency of knowledge-based mistakes is associated with a higher frequency of errors caused by the lack of skills. | Both knowledge and skills are required to provide high-quality and safe patient care in the clinical environment. | Knowledge-based mistakes are addressed during the debriefing part. Skills could be practised as a part of a simulation scenario or effectively as skill stations on simulators. |
| A higher frequency of lapses is associated with a higher frequency of errors caused by a sloth. | Lapses—errors that occur between the creation and execution of a plan—are among the most common in clinical practice. In contrast, sloth, characterized by a reluctance to perform tasks due to the effort required or a (perceived) imbalance between effort and reward, occurs less frequently. This concept, often associated with 'working on autopilot,' may arise from remaining within a comfort zone, a state in which lapses are also more likely to occur.. | A well-structured briefing and debriefing process, which establishes shared ground rules and fosters a safe learning environment, typically motivates participants to engage actively and focus on the assigned task. This approach can also serve as a model for delivering quality care in clinical settings. |
| A higher frequency of knowledge-based mistakes is associated with a lower frequency of errors caused by bravado. | Ideally, clinicians acquire competence based on their knowledge and expertise. Hence, errors caused by clinicians working beyond their competence or without adequate supervision (bravado) are less likely in more experienced staff and vice versa. | In the simulation, participants are encouraged to call for help early, speak up, and voice their opinion whenever they feel the task is beyond their competence or that an action could lead to patient harm. |
| According to tutors, a higher frequency of slips is associated with a lower frequency of errors caused by poor teamwork. | A slip is defined as an error caused by a lack of concentration. Good teamwork keeps participants focused on the situation, thus preventing slips. | Situational awareness is one of the most essential aspects of Crisis Resource Management, often taught during simulation courses. |

**Fig 7. Identified correlations between Reason's and Round's error taxonomies, with proposed interpretations and simulation training strategies.**

could generate additional data to validate our hypothesis that an optimal taxonomy for simulation-based medical education may involve a combination of these classification systems. This forward-looking approach is aligned with the broader goal of refining medical error taxonomies for enhanced applicability in simulation training.

## Strengths of the study

The key strength of our study lies in its innovative focus on bridging generic and domain-specific medical error taxonomies. By leveraging data from high-fidelity simulation sessions, meaningful associations with practical implications for healthcare education and training were identified. The inclusion of both participant and tutor perspectives enriched the dataset, providing a comprehensive view of error occurrences and their implications for simulation-based learning. Moreover, this study highlighted the educational value of simulated environments that serve as safe spaces for healthcare professionals to make mistakes. By fostering a non-punitive culture that views errors as learning opportunities, simulation training promotes the adoption of error management strategies and strengthens the overall safety culture in healthcare settings.

## Limitations of the study

Although this study offers valuable insights, it has some limitations must be acknowledged. First, the participants' and tutors' experiences reflected a specific subset of healthcare professionals involved in postgraduate simulation courses, which may not be representative of the broader medical community. Therefore, these findings warrant cautious generalization. Second, our reliance on survey data introduces the possibility of subjective bias, as responses may have been influenced by personal perceptions or the safe learning environment of the simulation center and the particular simulation scenario. Additionally, the exploratory nature of the study precludes definitive conclusions on the suitability of the proposed taxonomy blend for simulation-based medical education.

Future research should aim to develop and validate a comprehensive medical error taxonomy specifically tailored to simulation-based medical education. This would involve the collection of new datasets to ensure the reliability and applicability of the taxonomy across diverse clinical and educational contexts. Such efforts will address the current gap and provide a standardized framework for analyzing and mitigating medical errors.

## Conclusion

This study addresses a knowledge gap in medical error taxonomies for simulation-based training in medical fields related to Critical Care. By evaluating and identifying the correlations between generic and domain-specific taxonomies, we propose a blended framework that meets the unique needs of simulation education. Our findings emphasize the importance of simulation as a tool for understanding, managing, and preventing medical errors, thereby enhancing the overall healthcare performance regarding patient safety. Although further validation is required, the integration of Reason's and Round's medical error classification systems presents a promising direction for improving error management and fostering a culture of safety in healthcare.

## Supporting information

**S1 Appendix. Questionnaire employed for data collection in this study.**
(DOCX)

## Acknowledgments

We would like to thank Editage (www.editage.com) for English language editing.

## Author Contributions

**Conceptualization:** Tamara Skrisovska.

**Data curation:** Tamara Skrisovska, Daniel Schwarz.

**Formal analysis:** Daniel Schwarz.

**Funding acquisition:** Petr Stourac.

**Investigation:** Martina Kosinova.

**Methodology:** Tamara Skrisovska, Daniel Schwarz.

**Supervision:** Petr Stourac.

**Validation:** Petr Stourac.

**Visualization:** Daniel Schwarz.

**Writing – original draft:** Tamara Skrisovska, Daniel Schwarz.

**Writing – review & editing:** Daniel Schwarz, Martina Kosinova.

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
