## [Decision Letter · Decision Letter 0]

15 Jul 2024

PONE-D-24-13204Exploring Medical Error Taxonomies and Human Factors in Simulation-based Healthcare EducationPLOS ONE

Dear Dr. Schwarz,

Thank you for submitting your manuscript to PLOS ONE. After careful consideration, we feel that it has merit but does not fully meet PLOS ONE’s publication criteria as it currently stands. Therefore, we invite you to submit a revised version of the manuscript that addresses the points raised during the review process. Please submit your revised manuscript by Aug 29 2024 11:59PM. If you will need more time than this to complete your revisions, please reply to this message or contact the journal office at plosone@plos.org. Please include the following items when submitting your revised manuscript:A rebuttal letter that responds to each point raised by the academic editor and reviewer(s). You should upload this letter as a separate file labeled 'Response to Reviewers'.A marked-up copy of your manuscript that highlights changes made to the original version. You should upload this as a separate file labeled 'Revised Manuscript with Track Changes'.An unmarked version of your revised paper without tracked changes. You should upload this as a separate file labeled 'Manuscript'.

We look forward to receiving your revised manuscript.

Kind regards,

Mukhtiar Baig, Ph.D.

Academic Editor

PLOS ONE

Journal Requirements:

"This research was partially supported by the Specific University Research grant provided to  Masaryk University by the Ministry of Education of the Czech Republic (MUNI/A/1595/2023, MUNI/A/1551/2023) and also in part supported by the Ministry of Health of the Czech Republic (FNBr, 65269705). We thank Mr. Radomír Beneš and the American Manuscript Editors for the English language editing that greatly improved the quality of this manuscript."

"This research was partially supported by the Specific University Research grant provided to Masaryk University by the Ministry of Education of the Czech Republic (MUNI/A/1595/2023, MUNI/A/1551/2023) and also in part supported by the Ministry of Health of the Czech Republic (FNBr, 65269705)."

"This research was partially supported by the Specific University Research grant provided to Masaryk University by the Ministry of Education of the Czech Republic (MUNI/A/1595/2023, MUNI/A/1551/2023) and also in part supported by the Ministry of Health of the Czech Republic (FNBr, 65269705)."

5. Thank you for stating in your Funding Statement: 

"This research was partially supported by the Specific University Research grant provided to Masaryk University by the Ministry of Education of the Czech Republic (MUNI/A/1595/2023, MUNI/A/1551/2023) and also in part supported by the Ministry of Health of the Czech Republic (FNBr, 65269705)."

Reviewers' comments:

Reviewer's Responses to Questions

**Comments to the Author**

1. Is the manuscript technically sound, and do the data support the conclusions?

Reviewer #1: Partly

Reviewer #2: Partly

Reviewer #3: Yes

2. Has the statistical analysis been performed appropriately and rigorously? 

Reviewer #1: No

Reviewer #2: Yes

Reviewer #3: Yes

3. Have the authors made all data underlying the findings in their manuscript fully available?

Reviewer #1: No

Reviewer #2: No

Reviewer #3: Yes

4. Is the manuscript presented in an intelligible fashion and written in standard English?

Reviewer #1: Yes

Reviewer #2: Yes

Reviewer #3: Yes

5. Review Comments to the Author

Reviewer #1: i. Literature review is shallow

ii. Future scope is missing

iii. All abbreviations should only be used after their first definition

iv. Section should present the methodology algorithmically

iv. Novelty of the proposed work should be established by comparing the same with comparable work.

v. Sections break directly into sub-sections breaking the continuity of the manuscript.

vi. Sound literature review in tabulated form would be desirable to perform meta-analysis of available work and establish.

vii. There are areas that require further practical discussion and better linking of the results discussion together. The research methodology is weak. Why did you choose this methodology? The technical and practical discussion, as well as the comparison with recent previous work on this topic, should be thoroughly considered.

viii. The quality of the paper is weak in the technical discussion, and the explanatory results have been discussed before. Also, some parts in the comparison of figures don’t make sense. The results are not sufficient, and the conclusion is weak. The references are not sufficient and need a comprehensive update, as well as an update to the sources list until 2024. Major revisions are recommended so that the authors address all of this through examination and validation adequately to meet the standards and strength of the journal. The technical and practical discussion, as well as the comparison with recent previous work on this topic, should be thoroughly considered.

ix. Most sections have written with an AI tools and this missed sense in the information and understanding.

x. Both the discussion and practical application and the technical sense are missing. There is no discussion that conveys a true understanding.

xi. The introduction may provide a solid background on the importance of addressing medical errors and patient safety.

xii. Consider expanding on why existing taxonomies are inadequate for simulation-based settings upfront to justify the need for your proposed approach.

xiii. It is clearly stated that a p-value of under 0.1 and a correlation coefficient (tau) absolute value exceeding 0.1 were used as thresholds. However, the rationale behind choosing these thresholds could be elaborated. Provide a brief explanation for selecting these specific p-value and correlation coefficient thresholds. Example: “A p-value threshold of 0.1 was chosen to capture marginally significant associations, while a tau value exceeding 0.1 was used to identify moderately strong relationships.

xiv. Challenges related to error reporting are well-articulated. However, it would be beneficial to propose some strategies for overcoming these barriers.

xv. Suggest methods for mitigating gaps in literature review and enhancing future taxonomy validation. Example: “Future research should include a systematic review of less widely known taxonomies and consider the application of natural language processing to identify relevant literature.”

xvi. The manuscript discusses classifications and their relationships, but there is insufficient analysis on the accuracy and reliability of these classifications. The methodology for evaluating the accuracy of the proposed or merged classifications isn’t explicitly detailed.

xvii. The accuracy of the proposed classifications needs to be rigorously analyzed. Include metrics such as precision, recall, and F1 score to evaluate the performance of the classifications. Benchmark these against existing classifications to demonstrate their relative effectiveness.

xviii. The technical discussion is not currently high-level and lacks depth. Provide detailed descriptions of the methodologies and tools used to develop and implement the classification system in the simulation environment. Include discussions of any technical challenges encountered and how they were addressed.

xix. The proposed classification system needs stronger validation. Conduct validation studies in real-world simulation settings and present detailed results and analysis to support the efficacy of the modified taxonomy. Discuss how the findings from these validation studies enhance the credibility of your proposed approach.

xx. Sections of the manuscript involving AI are not well integrated and lack relevance. Clearly explain how AI technologies were utilized in the study, specifying the AI methods used and their role in the classification process. Highlight the impact of AI on the accuracy, efficiency, or reliability of the classification system with specific examples.

xxi. The paper needs proofreading to correct any typographical, grammatical, or structural errors. Ensure that terms and abbreviations are defined upon first use and employed consistently throughout the manuscript. Address any incomplete or unclear phrases, such as “human limitatUnderstanding.

xxii. Eaborate on how the study’s findings will be practically applied in simulation training. Provide specific examples of scenarios or training modules where the insights will be implemented. Detail how “red flags” will be incorporated into debriefing sessions.

Reviewer #2: Please include tables and charts for quantitative data collected through questionnaire. Please clarify how the study could be replicated in both national and international context. Also incorporate the suggested changes in the research manuscript, highlighted with comments attached in comment box.

Please refer to the reviewed manuscript for reference.

Reviewer #3: The methodology of the manuscript is robust, comprehensive and well-structured and thorough to identify critical gaps in literature. The topic is extremely crucial to the progress of medical education.

However, Methodology can be improved by clearly stating the study design of this manuscript in the beginning of the section. As in first glance it appears to be a review.

It would be good to clarify the various taxonomies and their usage in both types of medical errors with clarity instead of just a table.

The number of participants should be stated in methodology section instead of statistical analysis.

The level of the learner should be clearly specified. Is the study conducted on undergraduates, interns or residents?

Discussion should be more detailed and comprehensive, comparing the study results with previous studies.

A lot of material elaborated in the limitation's sections can be a part of discussion

Last paragraph of limitations is way forward, it should be separated from limitations

Literature is comprehensive and thoroughly cited, however the manuscript can be improved by citing latest research.

6. PLOS authors have the option to publish the peer review history of their article (what does this mean?). If published, this will include your full peer review and any attached files.

Reviewer #1: **Yes: **Ebrahim E. Elsayed

Reviewer #2: No

Reviewer #3: No

---

## [Author Response · Author response to Decision Letter 0]

13 Dec 2024

We have provided a rebuttal letter with detailed responses to all reviewers' comments.

---

## [Editor Report · Decision Letter 1]

22 Dec 2024

Exploring Medical Error Taxonomies and Human Factors in Simulation-based Healthcare Education

PONE-D-24-13204R1

Dear Dr. Schwarz,

We’re pleased to inform you that your manuscript has been judged scientifically suitable for publication and will be formally accepted for publication once it meets all outstanding technical requirements.

Kind regards,

Mukhtiar Baig, Ph.D.

Academic Editor

PLOS ONE

---

## [Editor Report · Acceptance letter]

10 Jan 2025

PONE-D-24-13204R1 

PLOS ONE

Dear Dr. Schwarz, 

I'm pleased to inform you that your manuscript has been deemed suitable for publication in PLOS ONE. Congratulations! Your manuscript is now being handed over to our production team.

Kind regards, 

on behalf of

Professor Mukhtiar Baig 

Academic Editor

PLOS ONE